# Poly-omic risk scores predict inflammatory bowel disease diagnosis

Christopher H. Arehart,[1,2,3] John D. Sterrett,[1,4] Rosanna L. Garris,[1,5] Ruth E. Quispe-Pilco,[1,2] Christopher R. Gignoux,[6] Luke M. Evans,[2,3] Maggie A. Stanislawski[6]

**ABSTRACT** Inflammatory bowel disease (IBD) is characterized by complex etiology and a disrupted colonic ecosystem. We provide a framework for the analysis of multi-omic data, which we apply to study the gut ecosystem in IBD. Specifically, we train and validate models using data on the metagenome, metatranscriptome, virome, and metabolome from the Human Microbiome Project 2 IBD multi-omic database, with 1,785 repeated samples from 130 individuals (103 cases and 27 controls). After splitting the participants into training and testing groups, we used mixed-effects least absolute shrinkage and selection operator regression to select features for each omic. These features, with demographic covariates, were used to generate separate single-omic prediction scores. All four single-omic scores were then combined into a final regression to assess the relative importance of the individual omics and the predictive benefits when considered together. We identified several species, pathways, and metabolites known to be associated with IBD risk, and we explored the connections between data sets. Individually, metabolomic and viromic scores were more predictive than metagenomics or metatranscriptomics, and when all four scores were combined, we predicted disease diagnosis with a Nagelkerke's $R^2$ of 0.46 and an area under the curve of 0.80 (95% confidence interval: 0.63, 0.98). Our work supports that some single-omic models for complex traits are more predictive than others, that incorporating multiple omic data sets may improve prediction, and that each omic data type provides a combination of unique and redundant information. This modeling framework can be extended to other complex traits and multi-omic data sets.

**IMPORTANCE** Complex traits are characterized by many biological and environmental factors, such that multi-omic data sets are well-positioned to help us understand their underlying etiologies. We applied a prediction framework across multiple omics (metagenomics, metatranscriptomics, metabolomics, and viromics) from the gut ecosystem to predict inflammatory bowel disease (IBD) diagnosis. The predicted scores from our models highlighted key features and allowed us to compare the relative utility of each omic data set in single-omic versus multi-omic models. Our results emphasized the importance of metabolomics and viromics over metagenomics and metatranscriptomics for predicting IBD status. The greater predictive capability of metabolomics and viromics is likely because these omics serve as markers of lifestyle factors such as diet. This study provides a modeling framework for multi-omic data, and our results show the utility of combining multiple omic data types to disentangle complex disease etiologies and biological signatures.

**KEYWORDS** omics, inflammatory bowel disease, gut microbiome, metabolomics, metatranscriptomics, viromics, multi-omics

Address correspondence to John D. Sterrett, john.sterrett@colorado.edu.

Christopher H. Arehart and John D. Sterrett contributed equally to this article. Co-first author order was determined by coin flip.

Luke M. Evans and Maggie A. Stanislawski are joint senior authors.

C.R.G. has stock in 23andMe.

See the funding table on p. 15.

Inflammatory bowel disease (IBD) is characterized by complex etiology and contains multiple pathological subtypes, including Crohn's disease (CD) and ulcerative colitis (UC). Both of these subtypes have wide-ranging heritability estimates and involve disruptions of the gut mucosa and dysbiotic gut microbiota. Despite clustering of these diseases within family trees, genome-wide association studies have produced much smaller single nucleotide polymorphism (SNP) heritability estimates (0.37 for CD and 0.27 for UC) (1) than twin studies, which estimate heritability coefficients of 0.75 for CD and 0.67 for UC (2). IBD affects individuals worldwide and has an incidence of 19.2/100,000 person-years in North America (3) and age-adjusted prevalence of 0.40% and 0.65% for CD and UC, respectively (4). With impactful symptoms including diarrhea, abdominal pain, bloody stools, weight loss, and fatigue, the rising rates of CD and IBD are a salient public health concern (5).

In genome-wide association studies to date, more than 120 related genes have been identified for CD, 67 of which were found to be differentially expressed for both CD and UC patients compared to non-IBD individuals (6). These biological underpinnings have connected IBD to an array of comorbidities including asthma (7) and diabetes (8), which are also characterized by polygenic architectures and complex environmental interactions related to immune activation. Additionally, many of the genes most strongly associated with IBD diagnosis are active in host-microbe interactions (such as pathogen-associated molecular pattern recognition, inflammatory responses, and phagocytic processes), including toll-like receptor (TLR) 4, TLR9, nucleotide-binding oligomerization domain containing 2, interleukin-23 receptor, and tumor necrosis factor (6). Multiple environmental risk factors for IBD have been identified, including smoking, excess body fat (9), urban living (10), antibiotic exposure (10), and adverse childhood events (11). On the other hand, studies have highlighted protective environmental factors including *Helicobacter pylori* infection (12), breastfeeding (9), and adequate vitamin D status (10). Notably, many of these environmental factors are associated with changes to an individual's gut microbiome, its activity, the host's metabolome, and immune function (13, 14).

Our aims for this study were twofold: (i) develop a generalizable and interpretable modeling framework that can used to study multi-omic data sets for complex traits by evaluating the predictive contribution of each omic data type, and (ii) apply this framework to the gut ecosystem in individuals with and without IBD to contextualize important features across four omic data sets. To accomplish both of these objectives, we reached beyond genetic data and genome-wide association studies to further uncover risk factors for IBD. This included microorganisms (both bacteria and viruses), RNA transcripts of the microbiome, and small molecules (metabolites) present in the gut. Whereas most studies including omic analyses use univariate differential abundance methods (examining one feature at a time), our goal was to create an additive and predictive multivariable model that would incorporate features from multiple omic data types. Given that many omic data types are compositional (15, 16) (meaning that the data are proportions or relative abundance constrained by library size), a decrease in one feature will correspond to an increase in others. Thus, multivariable differential abundance analysis or models for predicting disease status based on omic data (as will be described later in this paper) may provide a unique alternative to univariate differential abundance methods.

Analogous methods to polygenic risk scores (17) have begun to target sources of biological variation beyond genotypes. A recent study in the University of California, Los Angeles (UCLA) Health Biobank found that methylation-based risk scores were considerably more accurate than their genetic variant-based counterpart when imputing diagnoses and lab tests in the electronic health record (18). This recent pairing of machine learning methods and biological data from multiple omic levels has opened the door to better-informed models that may have greater potential for disease prevention and personalized medicine applications, especially among complex traits. The microbiome, transcriptome, virome, and metabolome are all dynamic through

time for individuals, which differ from genetic variants, which remain constant from birth. Therefore, a multi-omic-based prediction appears to be a promising approach in the context of complex and dynamic disease etiologies. Although high-throughput technologies and data generation are on the rise, there are still common issues such as overfitting to training data sets, low interpretability, and the general lack of portability from one context to another (19). We aim to address these shortcomings by developing a generalizable and interpretable modeling framework that can be applied across the phenome and to multiple, different omics. In the present analysis, we designed a modular multi-omic framework that allows researchers to assess the relative contributions of each omic data type individually and then combine them together to assess overall predictive capability.

## MATERIALS AND METHODS

### Data acquisition

The Human Microbiome Project 2 (HMP2) (20) IBD multi-omic database contains 1,785 unique samples collected from 131 participants across five study sites, with metadata including each sample's participant ID, sex, race, antibiotic use, and site. The data used in our study were publicly available (accessed online [https://ibdmdb.org/]) and collected following approval by multiple institutional review boards, referenced in the flagship HMP2 paper (20). All sample collection and participant involvement were reported to follow the Declaration of Helsinki guidelines and federal regulations.

### Data set description

Each of the 1,785 samples was analyzed across up to eight different omic data types. In this study, we used feature count tables for samples across four of the present omic data types: metagenomics (MGN), metatranscriptomics (MTS), viromics (VRM), and metabolomics (MBL). MGN data consisted of taxonomic counts at the species level, whereas MTS data were functional profiles at the pathway level. For clarification, the term "sample" is used in this paper to denote one fecal specimen, which could have corresponding MGN, MTS, VRM, and MBL data, whereas "participant" is used to denote an individual who may have multiple samples throughout the duration of the study. MBL, MGN, MTS, and VRM data were collected on average 10.3, 3.3, 6.5, and 7.4 weeks apart per participant, respectively (see Fig. S1). Count tables for MGN and VRM were generated with MetaPhlAn2 and VirMap, respectively. MTS count tables were generated by summing pathway abundances as mapped by HUMAnN2. For MBL, four column methods [Carbon 18 (C18) positive, C18 negative, hydrophilic interaction liquid chromatography (HILIC) positive, and HILIC negative) coupled with mass spectrometry were used to isolate and detect metabolites. Only compounds successfully annotated with molecule names were included in analyses.

### Data processing and thresholding

Compositional data sets (MGN, MTS, and VRM) were normalized with center log-ratio transformation, and MBL was normalized using a $log_{10}$ transformation. We then removed highly sparse features (found in fewer than 5% of samples) from VRM. For MBL, we only included compounds present in >99% of samples. The difference in methods used for these two data types is due to variation in the sparsity of the data sets, as the majority of viruses were found in very few samples, whereas the majority of compounds from MBL were found in most samples. Additionally, we removed one of any two highly collinear features (Pearson's $\rho > 0.95$) from MBL and MTS data sets at random. After normalization, features with a standard deviation less than 1 were excluded from MGN and MTS, and features with a standard deviation less than 0.1 were excluded from VRM and MBL. The standard deviation threshold for each omic data type was chosen based on a histogram of sample variation in the data set and served to eliminate features with

minimal differences across samples. The exclusion of high-missingness and low-variance features resulted in the filtering of MGN from 578 to 237 features, MTS from 421 to 280 features, VRM from 239 to 9 features, and MBL from 596 to 269 features (see Table S1).

## Training and validation split

Represented in our sample set were 130 distinct individuals (one participant from the original study was excluded due to missing data). To avoid overfitting, samples from 30 participants (23% of individuals) were set aside to serve as a validation set for performing model predictions and evaluating the resultant accuracy, as well as for use in our multi-omic model. Because the multi-omic model required all four data types, the 30 participants who had the most samples with all four omic data types and the fewest samples with missing omics were chosen for the validation set (Fig. S2). We ensured that no samples from these individuals were used in the training process for any of the four omic layers. Any samples in the validation set with missing omics were excluded from the multi-omic model. No aspects of the experimental design indicate a bias for how many samples from a participant contain all four omics (based on any qualities of that participant), so this is assumed to represent a non-biased subset of the participants to the best of the researchers' knowledge from all available information on sample collection. The remaining training set contained 100 participants with a varying number of samples across layers. Of the 30 participants reserved for validation, 15 (499 samples) had CD, 3 (102 samples) had UC, and 12 (427 samples) were non-IBD controls. Of the 100 training participants, 50 (1,242 samples) had CD, 35 (932 samples) had UC, and 15 (520 samples) were non-IBD controls. Table S1 shows the breakdown of training and validation samples within each omic data type.

## Individual omic models

Features for each omic data type were selected by applying mixed-effects least absolute shrinkage and selection operator (LASSO) (21) logistic regression using the glmmLasso function in the R package glmmLasso V1.5.1 (22) using the following model, where $y$ denotes IBD diagnosis, $\mu$ denotes the intercept, $\beta_i$ denotes the effect sizes, $n$ denotes the number of omic features, $(1|x)$ indicates a random effect for variable $x$, and $\epsilon$ denotes the error:

$$y = \mu + \sum_{i=1}^{n} \beta_i \times \text{feature}_i + \beta_{n+1} \times \text{age} + \beta_{n+2} \times \text{sex} + \beta_{n+3} \times \text{race} + \\ \beta_{n+4} \times \text{antibiotic\_use} + (1 \mid \text{site}) + (1 \mid \text{participant\_ID}) + \epsilon . \tag{1}$$

The mixed-effects LASSO regression was run across a grid of lambda values, which control how parsimonious the models are (via penalizing additional features). A final LASSO regression was run using the lambda value at the elbow of the number of features retained by LASSO (Fig. S3). Using only the LASSO selected features, we generated a logistic mixed-effects model (R package lme4 V1.1–29) (23) following equation 1 and requiring that the important covariates of age, sex, race, antibiotic use, site, and participant ID were included. Thus, feature weights were estimated in a context that accounted for these potentially confounding variables. Analogous to the methods for computing polygenic risk scores from a genome-wide association study (24), the $n$ feature weights (excluding participant demographic information) were then multiplied by each $i$th samples' feature value and summed, that is, for sample $i$:

$$\text{score}_i = \beta_1 \times \text{feature}_{1,i} + \beta_2 \times \text{feature}_{2,i} + \cdots + \beta_n \times \text{feature}_{n,i} . \tag{2}$$

These raw scores were standardized to have a mean of 0 and a standard deviation of 1.

## Baseline models

We fit a baseline risk score model for each omic data type based on only metadata for each participant using the lmer function via the lme4 R package where terms are the same as described above:

$$y = \mu + \beta_1 \times \text{age} + \beta_2 \times \text{sex} + \beta_3 \times \text{race} +$$
$$\beta_4 \times \text{antibiotic\_use} + (1 \mid \text{site}) + (1 \mid \text{participant\_ID}) + \epsilon. \tag{3}$$

We trained these models on participants in the training data set and then used them to predict the diagnosis of participants in the validation data set.

## Assessing model validity

Within each omic data type, predicted risk scores for each individual were averaged across their longitudinal samples in the validation set (see Fig. S4). Area under the curve (AUC) values were calculated from receiver operating characteristic curves for each model (both baseline and feature) across omic layers. Odds ratios (ORs) were also calculated for each of the predicted scores, and 95% confidence intervals (CIs) were generated for each OR.

## Multi-omic models

Prediction scores generated via feature weights within each omic layer were averaged across samples with all four omics present for each individual in the testing data set and then combined in the following logistic regression model:

$$y = \mu + \beta_1 \times \text{MGN} + \beta_2 \times \text{MTS} + \beta_3 \times \text{VRM} +$$
$$\beta_4 \times \text{MBL} + \beta_5 \times \text{age} + \beta_6 \times \text{sex} + \epsilon. \tag{4}$$

Age and sex were the two covariates included in this model due to little or no variability across the validation samples for other variables such as race or antibiotic use. Nagelkerke's $R^2$ was used to compare the combined logistic model to a baseline model only based on age and sex. Code for these methods has been made publicly available at the following repository (https://github.com/sterrettJD/poly-omics-risk).

## Analyzing metabolite origins

We used Annotation of Metabolite Origins using Networks (AMON) (25) to link the MBL data set to the MGN data set. AMON uses the Kyoto Encyclopedia of Genes and Genomes (KEGG) (26) Orthologies (KOs) present from microbial taxa and the host genome to resolve which KEGG compounds were more likely derived from microbial sources, the host, or other sources. We mapped the names of annotated, LASSO-selected compounds to KEGG compound identifiers, which involved collapsing duplicated names to single compound identifiers. A list of KOs present in the human genome was downloaded directly from KEGG, using the "hsa" organism code. The list of microbial KOs was derived from the publicly available functional profile of the MGN data set generated using the HUMAnN pipeline.

## RESULTS

### Included participants

This study included 130 participants, of whom, 27 did not have IBD, 65 had CD, and 38 had UC. Male and female participants were represented similarly, and the mean age of participants was 28 years old, with a standard deviation of 17 years and a range of 6–76 years of age. The majority of participants were White and did not use antibiotics. Table 1 elaborates with descriptive statistics of each group.

**TABLE 1** Descriptive statistics of participants in the study[a]

|  | Healthy control (n = 27) | Crohn's disease (n = 65) | Ulcerative colitis (n = 38) | Total (N = 130) |
|---|---|---|---|---|
| Sex |  |  |  |  |
| Female | 12 (44.4%) | 32 (49.2%) | 20 (52.6%) | 64 (49.2%) |
| Male | 15 (55.6%) | 33 (50.8%) | 18 (47.4%) | 66 (50.8%) |
| Age |  |  |  |  |
| Mean (SD) | 29 (±20) | 26 (±16) | 29 (±17) | 28 (±17) |
| Antibiotics |  |  |  |  |
| No | 27 (100%) | 51 (78.5%) | 33 (86.8%) | 111 (85.4%) |
| Yes | 0 (0%) | 14 (21.5%) | 5 (13.2%) | 19 (14.6%) |
| Site |  |  |  |  |
| Cedars-Sinai | 1 (3.7%) | 20 (30.8%) | 12 (31.6%) | 33 (25.4%) |
| Cincinnati | 9 (33.3%) | 17 (26.2%) | 7 (18.4%) | 33 (25.4%) |
| MGH | 13 (48.1%) | 13 (20.0%) | 11 (28.9%) | 37 (28.5%) |
| MGH Pediatrics | 3 (11.1%) | 8 (12.3%) | 5 (13.2%) | 16 (12.3%) |
| Emory | 1 (3.7%) | 7 (10.8%) | 3 (7.9%) | 11 (8.5%) |
| Race |  |  |  |  |
| American Indian or Alaska Native | 0 (0%) | 1 (1.5%) | 0 (0%) | 1 (0.8%) |
| Black or African American | 1 (3.7%) | 3 (4.6%) | 6 (15.8%) | 10 (7.7%) |
| More than one race | 1 (3.7%) | 3 (4.6%) | 1 (2.6%) | 5 (3.8%) |
| White | 25 (92.6%) | 56 (86.2%) | 29 (76.3%) | 110 (84.6%) |
| Others | 0 (0%) | 2 (3.1%) | 2 (5.3%) | 4 (3.1%) |

[a]Categorical data are presented as "N (percent)," and numerical data are presented as "mean (standard deviation)." Abbreviations: MGH, Massachusetts General Hospital; SD, standard deviation.

## Baseline models

Baseline models, which only include patient demographic information and were trained on the training data set, poorly predicted the actual diagnosis of our validation data set, as can be seen in Table S2 (MGN AUC [95% CI] = 0.43 [0.19, 0.67], MGN Nagelkerke's $R^2$ = 0.08; VRM AUC [95% CI] = 0.47 [0.23, 0.71], VRM Nagelkerke's $R^2$ = 0.02; MTS AUC [95% CI] = 0.38 [0.15, 0.62], MTS Nagelkerke's $R^2$ = 0.04; MBL AUC [95% CI] = 0.47 [0.25, 0.69], MBL Nagelkerke's $R^2$ = 0.01). This established that diagnosis could not be discriminated solely by participant metadata and non-omic covariates as described in equation 3 and Table S1.

## Individual omic risk scores

Compared to the baseline models, the predicted risk scores (using feature weights derived from the training set) for each omic data type (in addition to the baseline covariates of age and sex) demonstrated better predictive capability on the validation data as is shown in Fig. 1 (MGN AUC [95% CI] = 0.66 [0.44, 0.87], MGN Nagelkerke's $R^2$ = 0.12; VRM AUC [95% CI] = 0.83 [0.68, 0.98], VRM Nagelkerke's $R^2$ = 0.43; MTS AUC [95% CI] = 0.73 [0.53, 0.92], MTS Nagelkerke's $R^2$ = 0.20; MBL AUC [95% CI] = 0.82 [0.66, 0.98], MBL Nagelkerke's $R^2$ = 0.40). A similar plot showing AUCs and ORs when only using the omic-derived scores to predict diagnosis is shown in Fig. S5 with similar results, suggesting that the covariates of age and sex are not driving this predictive accuracy (MGN AUC [95% CI] = 0.70 [0.50, 0.91], MGN Nagelkerke's $R^2$ = 0.04; VRM AUC [95% CI] = 0.80 [0.63, 0.96], VRM Nagelkerke's $R^2$ = 0.31; MTS AUC [95% CI] = 0.64 [0.42, 0.86], MTS Nagelkerke's $R^2$ = 0.12; MBL AUC [95% CI] = 0.78 [0.59, 0.98]), MBL Nagelkerke's $R^2$ = 0.32]. Of the individual omic models, VRM and MBL had the best predictive capability with little to no interquartile overlap between cases and controls. The VRM and MBL predicted scores had significant ORs of 9.28 (1.53, 56.13; $P$ = 0.02) and 4.30 (1.32, 13.98; $P$ = 0.02),

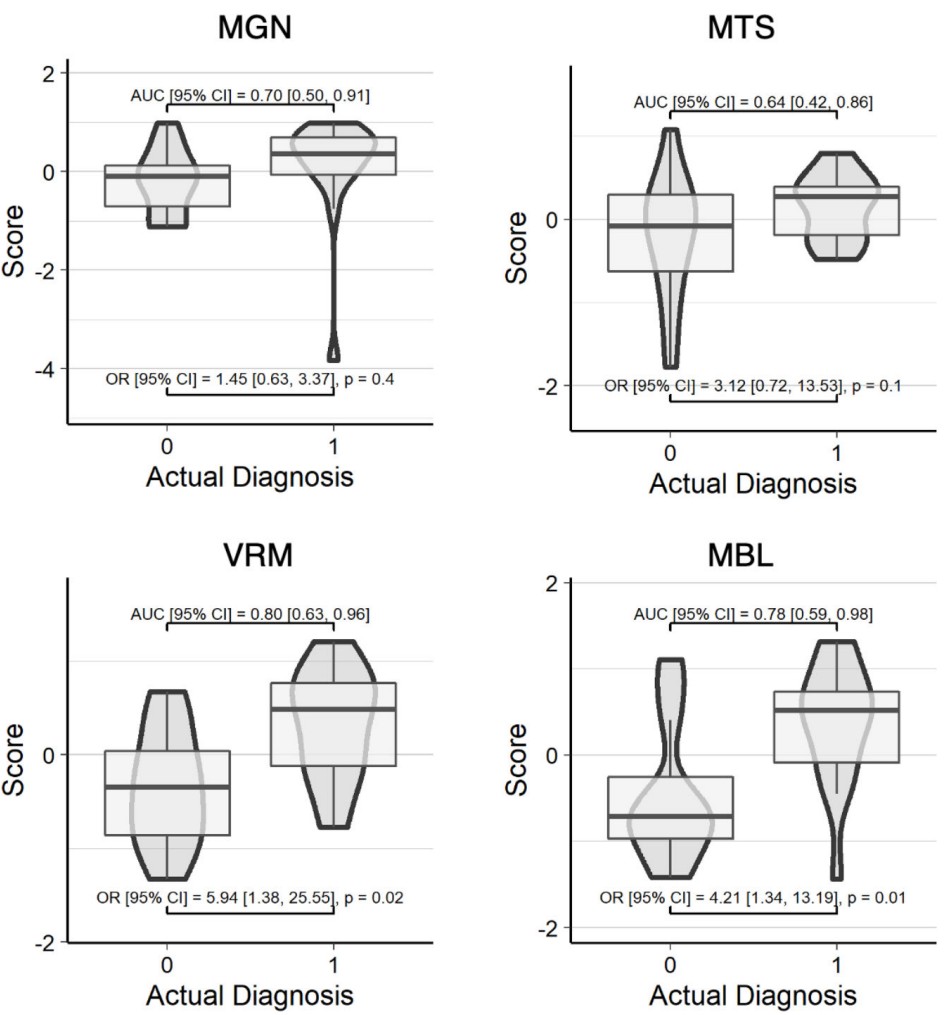

**FIG 1** Risk scores predict IBD diagnosis. *Z*-score transformed risk scores (averaged across all samples for each participant) on the *y* axis are plotted against actual diagnosis on the *x* axis for the validation data set. AUC and OR were calculated with basic covariates (diagnosis ~ score + age + sex). Each of the four scores shown was calculated using feature weights from a LASSO-identified mixed-effects logistic regression trained in a separate set of samples/individuals: diagnosis ~ features + age + sex + race + antibiotic use + (1|site) + (1|participant ID). An actual diagnosis value of 1 indicates the presence of IBD.

respectively, while MTS had an OR of 2.89 (0.62, 13.52; $P = 0.2$), and MGN had the lowest OR of 1.24 (0.50, 3.09; ($P = 0.6$).

## Selected features

Within each individual omic model, LASSO selected 14 species from the 237 considered in the MGN data set, representing 5.91% of all features passing quality control (QC); 23 pathways from the 280 considered in the MTS, representing 8.21% of all features passing QC, 14 features from 269 metabolites in the MTB, representing 5.20% of all features passing QC; and six viruses from nine considered in the VRM, representing 66.67% of all features passing QC. Figure 2 shows the feature weights for each omic data type. For MGN, *Megasphaera* sp. DISK18 had the strongest negative weight (indicating association with a low risk score). Additional species associated with lower risk scores included *Parabacteroides goldsteinii*, *Methanobrevibacter smithii*, *Roseburia hominis*, and *Akkermansia muciniphila*. *Firmicutes* CAG 83 was the only species identified with a positive weight. Figure S6 to S19 illustrate the dynamic longitudinal trends among the 14 LASSO-selected MGN species for the 30 individuals reserved for model validation. The variation and volatility within individuals over the 52-week sampling period highlight the importance

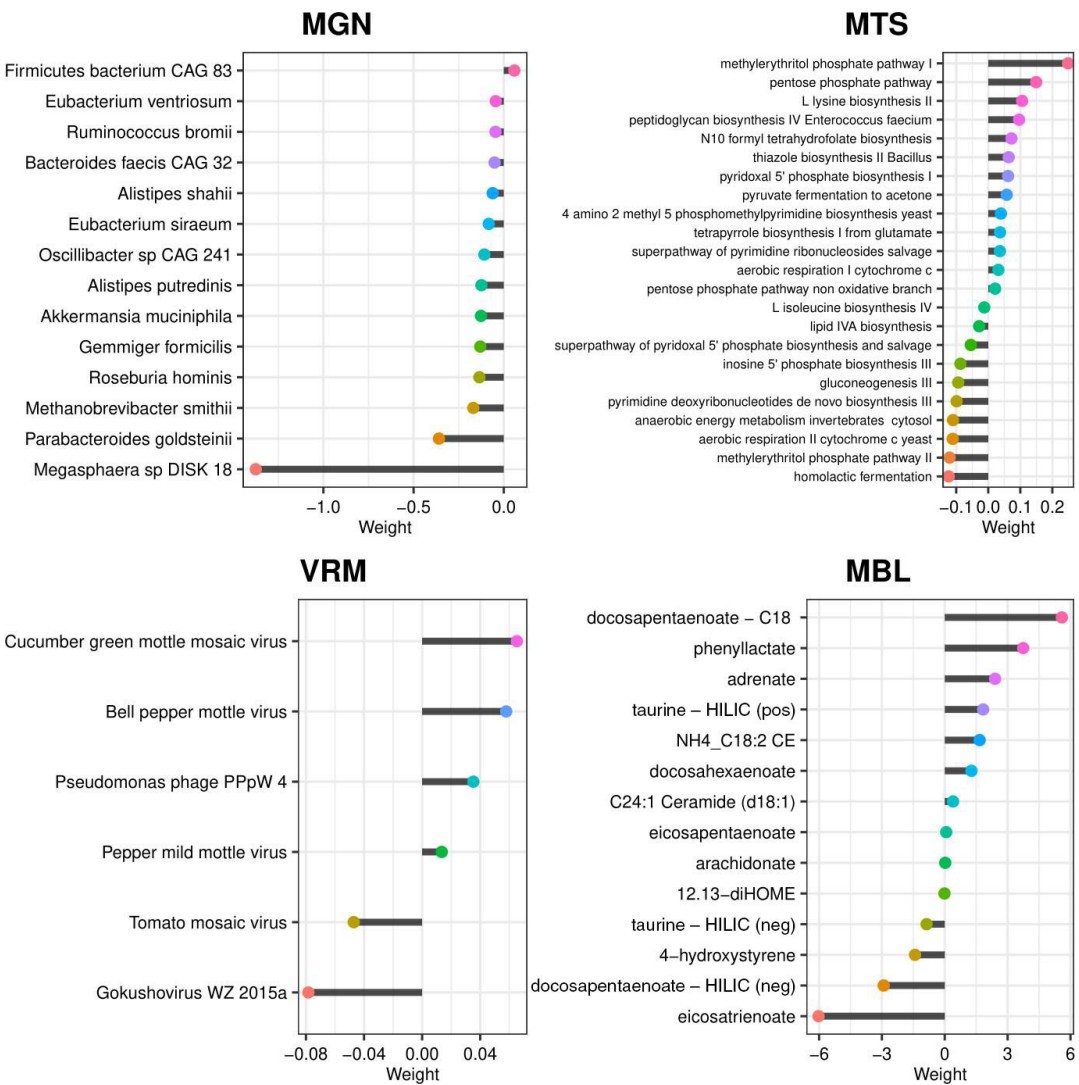

**FIG 2** Feature weights for each risk score model. The effect sizes described by the x axis were multiplied by transformed abundances and then summed to generate each omic score. A negative weight corresponds to a lower predicted risk of IBD, whereas a positive weight would confer a higher predicted risk of IBD.

of longitudinal sampling when trying to capture robust estimates of the effects of each taxon. Additionally, a phylogenetic tree of the MGN features shows that *Alistipes putredinis* and *Alistipes shahii* are the most related species, while *M. smithii* appeared to be more distantly related to the rest of the features (Fig. S20). A co-occurrence analysis revealed that the 14 selected features showed low levels of co-occurrence (<0.5 Pearson correlation index). Among the 14 features, the pairs *Eubacterium siraeum* and *Oscillibacter* sp. CAG:241, as well as *A. putredinis* and *A. shahii*, had the highest Pearson index of 0.5. *R. hominis* and *Megasphaera* sp. DISK_18 had the lowest Pearson index of −0.03 (Fig. S21 and S22). The absence of large negative values across the heatmap suggests that these 14 features do not have mutually exclusive relationships.

The MTS model selected 22 pathways. Of these, the strongest negative weights (associated with a lower risk score) were for the 2-methyl citrate cycle I and the superpathway of sulfur amino acid biosynthesis identified from *Saccharomyces cerevisiae*, and the strongest negative weights were for the pentose phosphate pathway and the methylerythritol phosphate pathway I. Of the six viruses included in the VRM model, four had positive weights indicating a direct association with IBD. The strongest positive weight was for cucumber green mottle mosaic virus and the strongest negative weight

was a gokushovirus. The MBL model selected 14 metabolites. The strongest positive weights were for docosapentaenoate (identified by the C18 column) and phenyllactate, and the strongest negative weights were for eicosatrienoate and docosapentaenoate (identified by the HILIC column).

## Combined multi-omics model

Figure 3a illustrates the ORs of all omic scores in a combined framework for the 30 individuals reserved for model validation. The covariates of age and sex were only slightly predictive of IBD on their own (Nagelkerke's pseudo-$R^2$ = 0.11); however, a multiple regression including age, sex, and all four scores produced a Nagelkerke's $R^2$ = 0.46 and an AUC of 0.80 (Fig. 3c). The variance explained by the four scores without age and sex was slightly lower ($R^2$ = 0.37), showing the importance of including demographic covariates and the predictive capacity of our combined framework to explain variance among IBD diagnoses. Despite the 95% CI for all four scores of ORs overlapping with one, VRM and MBL stood out with larger values than MGN or MTS. The correlation between scores illustrates some similarity between VRM, MGN, and MBL, in addition to MTS and MBL (Fig. 3b). However, these correlations are modest in magnitude (<0.5), suggesting that each of these scores captures a mostly distinct signal. Analysis for a

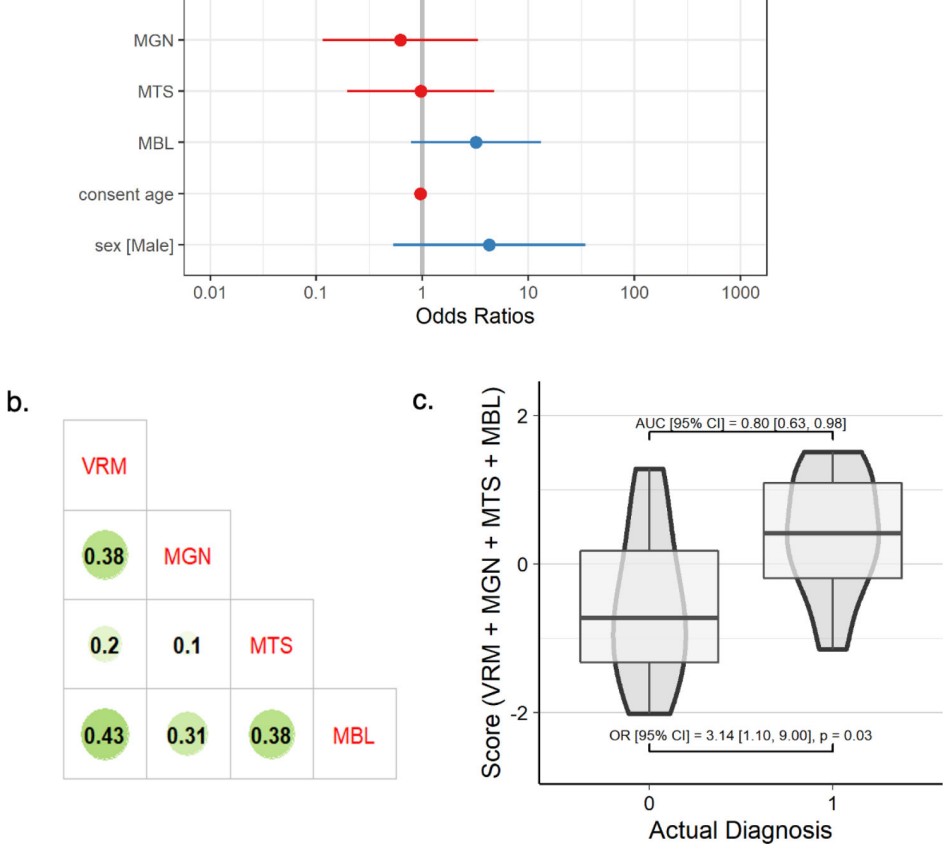

**FIG 3** Results of multi-omic modeling. (a) ORs and 95% CIs of the predicted score, (b) Pearson correlation matrix between omic scores, and (c) standardized risk scores (summed across all omics) on the $y$ axis plotted against actual diagnosis on the $x$ axis for the validation data set, where AUC and OR were calculated with basic covariates (diagnosis ~ score + age + sex). In panel a, points represent the OR for each omic's predicted scores in the multi-omic regression, and lines represent 95% CIs of the ORs. In panel b, the size and darkness of the circle represent the correlation between the predicted scores for each omic data type. In panel c, an actual diagnosis of 1 represents a case for IBD.

leave-one–omic-out approach is shown in Fig. S23 to further demonstrate the increased ORs (near the threshold for significance) for VRM and MBL compared to MGN and MTS.

## Metabolite origins

AMON (25) was used to predict the origins of the 14 metabolites selected by LASSO using the host (human) genome and the genomes of the 237 bacteria species in the MGN data set. Of the 14 selected metabolites, two had the same compound identifier, resulting in 12 compounds that were unique and annotated. We used a list of KOs in the human genome and from the KOs identified by the shotgun metagenome functional profile. AMON predicted that seven of the 12 MBL compounds were produced by either the human or the gut microbiome, meaning that five of the compounds were likely produced by other sources (such as plant compounds derived from dietary sources). Figure 4 shows the classification of these seven compounds based on their likely organismal source. Of the seven identified compounds, four were likely only produced by the human, two were produced by either the human or the microbes, and one was likely only produced by the microbes present in the gut microbiome. The LASSO-selected taxa contained KOs to produce two of the metabolites, taurine and 4-hydroxystyrene. Notably, only the LASSO-selected bacteria were capable of producing 4-hydroxystyrene, whereas the non-selected taxa and host were not.

## DISCUSSION

Overall, we successfully utilized a polygenic risk score framework across multiple omic data types to predict IBD diagnosis. Other studies have taken similar approaches, such as single variable differential abundance testing across multiple omics in IBD (27), multivariate analysis of multi-omic interactions in participants with IBD (28), like a composite of unsupervised multivariate analysis and principal component analysis, or multi-omic risk scores for diseases other than IBD (29). Our results often corroborate the findings of other multi-omic studies of individuals with IBD, and they also highlight groups of features that associate with IBD when considering the abundances of other features. Additionally, our methods allow for comparison of the predictive ability of

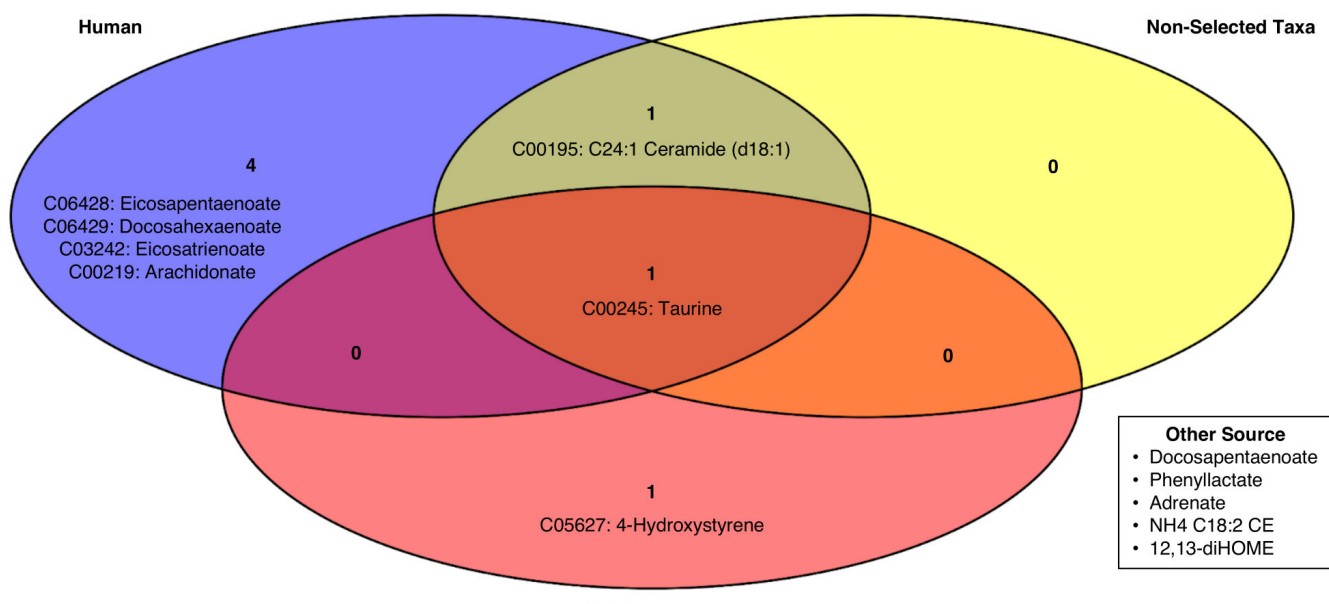

**FIG 4** Annotation of LASSO-selected compound origins. A Venn diagram showing the likely origins of LASSO-selected compounds from the MBL data set. "LASSO-selected taxa" refers to KOs from the 14 taxa selected by the MGN LASSO model, whereas "non-selected taxa" refers to all KOs detected in the MGN data set from the 223 taxa not selected by the MGN LASSO model. "Human" refers to all KOs present in the human genome, according to KEGG database, and "other sources" refers to compounds that were likely not produced by the host or detected taxa from the gut microbiome.

different omics for disease, highlighting that taxonomic profiling (from MGN) might not hold as much of a predictive signature of IBD as other omics that provide information on active transcription (MTS) and compound presence (MBL). Additionally, although VRM-predicted IBD scores correlated with MGN, VRM held a much stronger relationship with IBD, which could be a result of agricultural viruses serving as markers of diet.

## Interpretation of features identified by individual models

In the sections below, we explore the biological relevance of the LASSO-selected features for each of the omic models. This discussion of the selected features is meant to describe the potential biological relevance underlying each of the models. However, it is important to note that these conclusions are not focused on nominally significant features; instead, the LASSO-selected features are considered in their multivariable context, where many taxa or features could be interdependent (due to factors such as microbial cooperation/competition or mediation of the effects of one pathway by another). We describe features that were selected by LASSO for their joint predictive utility. These should not be confused with results from independent association tests between each feature and the outcome (such as differential expression analysis), which would generally be used to identify individual features significantly associated with the outcome after correction for multiple hypothesis testing. When applied to the testing data set, the combined multi-omic model was more predictive than the baseline model, and this is the focus of our results; the following sections aim to contextualize the underlying biological relevance of these models.

### Metagenomics

Among the 14 MGN features selected by LASSO, all but one had a negative weight in the multiple regression. These negative effect estimates correspond to reduced risk of IBD when utilized in the scoring framework, and the estimate with the largest magnitude (−1.37) was *Megasphaera* sp. DISK 18, which is known for being an early colonizer of the oral microbiome (30). The next most negative effect estimate, from *P. goldsteinii*, is of key interest given recent findings in mouse experiments that a *P. goldsteinii* probiotic may be useful in treating diet-induced obesity and type 2 diabetes (31). The links between the microbiome, diet, and IBD are important to consider in the context in which environmental exposures may manifest a genetic predisposition to the onset of IBD (8). Our findings of *R. hominis* and *Bacteroides faecis* CAG 32 associated with decreased risk of IBD are consistent with existing literature (32). We are intrigued by the importance of two features belonging to the *Alistipes* genus given the emerging connections between *Alistipes* and gut dysbiosis (33). Lastly, *Ruminococcus bromii*'s role in the MGN score is worth highlighting since *R. bromii* is known to support the growth of *Ruminococcus gnavus* (34), which is a species purported to be associated with increased risk of CD (32, 35).

### Metatranscriptomics

The MTS model identified a strong negative weight for transcripts belonging to the 2-methyl citrate cycle, meaning this pathway was associated with a lower risk of IBD. Notably, the 2-methyl citrate cycle is responsible for metabolizing propionic acid (36) (a short-chain fatty acid produced by gut commensal organisms fermenting dietary fiber) into succinate, which can enter the tricarboxylic acid cycle. Increased transcription of this pathway could be indicative of high propionic acid in the gut, which has previously been demonstrated to be protective in the context of IBD, as propionic acid has anti-inflammatory effects through the inhibition of nuclear factor κB (37). Alternatively, lower propionic acid could reflect decreased dietary fiber in participants with IBD. This result also aligns with the previous analysis (38) of the HMP2 data set (using different methods). The MTS model additionally identified a strong negative weight for sulfur amino acid biosynthesis transcripts. Given that one signature of IBD is a microbial shift toward the catabolism

of taurine and cysteine to produce hydrogen sulfide, this is consistent with previous literature as well (39).

The MTS model identified the pentose phosphate and methylerythritol phosphate pathways as having the strongest positive weights. The pentose phosphate pathway is involved in the metabolism of C5 sugars, but not much literature exists explaining a connection between it and IBD, although one study identified an increase in the pentose phosphate pathway in ileal CD (40). Similarly, little literature exists about the potential role of the methylerythritol phosphate pathway in the context of IBD, although its intermediates are potent activators of human gamma delta T cells, which are the first line of mucosal defense (41). Gamma delta T cells have been implicated in intestinal inflammation and IBD in both human and animal models across a multitude of studies, although it is unclear whether their effects are protective or not (42).

## Viromics

The VRM model identified six viruses associated (either positively or negatively) with IBD risk. Four of these, cucumber green mottle mosaic virus, bell pepper mottle virus, pepper mild mottle virus, and tomato mosaic virus, are all in the *Tobamovirus* genus. These are highly persistent and transmissible positive-sense single-stranded RNA viruses that are known to infect various crop species and are found worldwide. Although many crops have developed resistance to the damaging effects of these viruses, they are still endemic in agricultural products and have been found to make up a large portion of the gut virome (43, 44). As such, the presence of these viruses could serve as a marker of food choice in these participants, given that individuals with IBD may choose to consume different foods due to their condition. Interestingly, only one of these tobamoviruses, tomato mosaic virus, had a negative feature weight. The other two viruses identified, pseudomonas phage PPpW4 and gokushovirus WV 2015a, are both bacteriophages. Pseudomonas phage PPpW4 parasitizes pseudomonas bacteria, which are associated with pneumonia and post-surgical infections, and have been studied for use in phage therapy (45). Gokushoviruses, although ubiquitous throughout the environment, are largely uncharacterized (46).

## Metabolomics

The MBL model identified 14 metabolites associated with IBD, the majority of which were fatty acids. Metabolites with positive weights were docosahexaenoate, $NH_4$-18:2 cholesterol ester, C24:1-ceramide (d18:1), adrenate, phenyllactate, eicosapentaenoate, and arachidonate. Adrenate and phenyllactate have been found in increased concentrations in individuals with IBD (20). Metabolites with negative weights were hydroxystyrene, eicosatrienoate, and 12,13-dihydroxyoctadec-9-enoic acid (diHOME). 12,13-diHOME was also reported as an important metabolite for differentiating IBD status when applying a knockoff filtering-based multivariate approach to data from HMP2 (47).

Our model selected two compounds annotated as taurine that had been isolated using different chromatographic columns (HILIC negative and HILIC positive), and interestingly, these compounds had contrasting positive and negative weights. This may be related to the charge (or other aspects) of the identified compounds, but a more detailed investigation to follow up the untargeted MBL would be needed to differentiate these two. In accordance with this uncertainty, multiple studies have found conflicting associations between taurine in the gut metabolome and IBD status (48). Taurine, being a sulfur-containing amino acid, could have very diverse roles in the gut metabolome given its microbial metabolism to hydrogen sulfide, which can have effects ranging from pathogen inhibition at moderate concentrations to direct irritation of the gut mucosa at high concentrations (49). Additionally, taurine plays other roles in the gut metabolome, such as conjugating with cholic acid to form the bioactive secondary bile acid, taurocholic acid. Our findings are consistent with some differential abundance testing results from the original HMP2 paper, as they found taurine and taurocholic acid to be differentially abundant across IBD status (20).

Similarly, our model selected two compounds annotated as docosapentaenoate, although, in this case, one with a positive weight came from the C18 column (specialized for intermediate polarity compounds, such as free fatty acids), and one with a negative weight came from the negative ion mode HILIC-negative mode (specialized for polar compounds). According to previous studies, docosapentaenoate is an omega-3 fatty acid generally regarded to have anti-inflammatory effects (50).

## Comparing individual omic models

Of the individual omic models, MBL and VRM had the highest predictive capability, while MTS provided moderate prediction, and MGN provided the lowest predictive accuracy. One explanation for these results is that MBL and VRM provide information about both the microbiome and the host; the metabolites could have been produced by the host or their diet, and the viruses could also be diet-related given that four of the six selected viruses belong to the genus *Tobamovirus* and infect crops. In contrast, MTS and MGN only provide information about the microbiome, not the host. However, MTS specifies data on active transcription, whereas MGN only provides information about taxonomy and functional capabilities. Notably, our MGN analysis only considered taxonomy, not functional potential (i.e., no MGN pathway abundance data). These results are consistent with other findings that the metabolome predicts phenotypes better than taxonomic profiles do (51).

## Comparison of features selected by single-omic models

AMON identified that most of the LASSO-selected compounds from the MBL data set were not produced by the taxa selected by the MGN LASSO model. This suggests that these single-omic models are providing orthogonal information, which is complemented by the weak correlation between MGN and MBL single-omic scores. Of note, the compound 4-hydroxystyrene was identified to be produced by the MGN LASSO-selected taxa (but not the host or non-selected taxa). This has been demonstrated experimentally, as other groups have shown that microbial metabolism of hydroxycinnamic acids produces 4-hydroxystyrene in the rat microbiome (52). Particularly, polyphenols such as hydroxycinnamic acids (and their metabolites) have been studied as potential interventions in IBD (53), as they modulate epithelial inflammation and severity of dextran sulfate sodium-induced UC in mice (54). Thus, the integrated results of our multi-omic modeling highlight the importance of microbial metabolism of select polyphenols on IBD status and the relevance of a multi-omic modeling approach for this disease. However, with a larger sample size, an approach allowing for interactions between omic data sets may improve the detection of microbes involved in the metabolism of phenotype-modulating compounds.

## Combined model

The combined regression model incorporating multiple omic scores allows us to contextualize the predictive capability of each omic model in a multi-omic context. The results of our combined model mirror the AUCs and ORs of the individual omic models. In the combined model, MGN had the lowest OR, whereas MTS had a moderate OR, and MBL and VRM had the highest ORs. We were limited by our sample size to not be able to reserve a second bonafide validation set for the combined model, but the improved $R^2$ from a baseline model indicates the predictive capability of this combined model. Other studies performing multi-omic predictive modeling should consider similar frameworks with single-omic predictive scores to improve the interpretability of important features and allow for the comparison of predictive accuracy between various omic data types. This modular approach of building scores within each omic data type allows for the selection of a sparse set of features within each data set; it also leads to a combined model with insights contextualized by multiple biological omic layers (ideally ranging from the genome to the metabolome).

## Limitations

As with many studies of this kind, batch and site effects may affect the generalizability of our model. High-throughput omic technology faces many limitations due to batch effects, and future work should consider techniques for combining batches in this framework, using batch correction techniques, or incorporating random effects for batches (55). Such work to combine data from different studies could increase the robustness of this framework and increase the generalizability of its diagnostic capability, which would increase the likelihood of clinical implementation for such diagnostic omic modeling, which should be considered an end goal.

Additionally, the number of omic features was far greater than the number of participants, although we were able to use multiple longitudinal samples per participant (accounted for via random effects for participant ID). We took steps to reduce the number of features, such as through the sparsity, collinearity, and variance filtering steps and through the use of LASSO regression for feature selection. However, such steps could cause us to unwittingly discard useful information. Importantly, our modeling did not consider interactions between features, either within or between omics, which is a direction for future research. Moreover, the use of a center log-ratio transformation is not robust to shifting total microbial biomass, nor is it subcompositionally consistent. Other log-ratio transformations, such as the isometric log-ratio transformation, are subcompositionally consistent, although they do suffer from more difficult interpretability. Future work should consider the use of additive, isometric, or phylogenetic isometric log-ratio transformations to transform compositional data out of the simplex into Euclidean space, although interpretability and use cases of poly-omic models should be considered as a trade-off.

The relatively small sample size and sample overlap across omic data sets limited our ability to detect weak effect sizes across features. Extending these analyses to larger and more diverse studies would allow us to evaluate these methods on a larger scale. For example, performing cross-validation splits of the training and testing data sets would have further tested the robustness of these results. However, without sufficient sample sizes overlapping between omic layers, we were limited to one training-testing split of the data. In addition, exploring non-linear machine learning techniques such as random forests may have yielded better results and motivates an exciting direction for future multi-omic analyses.

The currently available data set from HMP2 does not allow public access of host genome data, which would have provided immense context to the "outer" omics upon which our analysis focused, and future analysis incorporating host genome data would add to these results by allowing comparison of the outer omic to standard polygenic risk scores for IBD. Additionally, no information was available regarding the existence of any relatives within the data set, which could have confounded results, as our analyses assumed all participants were not related or living in the same location.

## Conclusions

Not only did we successfully implement a poly-omic risk score framework across four omic layers, but also each of our individual models identified features known to associate with IBD risk while providing new insights into features that may influence or be influenced by IBD. The majority of IBD models do not utilize multiple omic layers, focusing instead on host genetics or solely microbiome taxonomic composition (56). Those that utilize other information often focus on predicting flare-up and other clinical outcomes among individuals with IBD (39). This study is unique in that it combines omic layers to predict the presence of disease.

Our results provide a framework for an interpretable comparison of single-omic models in multi-omic contexts, with particular relevance to the gut microbiome and complex phenotypes. Our work suggests that some single-omic models (in this case MBL and VRM) are more predictive than others, although this is likely phenotype-dependent. Each omic model provides a combination of unique and redundant information,

relative to other omic models, and a combination of single-omic models may often yield improvements in predictive accuracy. Such methods are extendable and customizable to the context of interest through the use of non-linear methods, alternative data transformations, and interactions between features. There are generally trade-offs between model simplicity, predictive accuracy, and interpretability, and future studies should carefully consider the primary goals of their model. Multi-omic modeling approaches that connect the dots along the continuum from genes to their environment will be paramount to identifying novel insights for health and disease.

## ACKNOWLEDGMENTS

We would like to thank Kristin Powell and Stephanie Hoyt for their support for this project, as well as Tom Cech and the BioFrontiers Institute for their support of the Interdisciplinary Quantitative Biology program. We would also like to thank Lukas Buecherl and Casey Martin for providing manuscript feedback and the Human Microbiome Project 2 team for data collection, processing, and public access and HMP2 participants.

This project was funded by the National Science Foundation-sponsored Interdisciplinary Quantitative Biology PhD program, the Integrated Data Science (Int dS) Graduate Training Fellowship, and the William J. Freytag Fellowship.

## AUTHOR AFFILIATIONS

[1]Interdisciplinary Quantitative Biology PhD Program, University of Colorado, Boulder, Colorado, USA

[2]Department of Ecology and Evolutionary Biology, University of Colorado, Boulder, Colorado, USA

[3]Institute for Behavioral Genetics, University of Colorado, Boulder, Colorado, USA

[4]Department of Integrative Physiology, University of Colorado, Boulder, Colorado, USA

[5]Department of Biochemistry, University of Colorado, Boulder, Colorado, USA

[6]Department of Biomedical Informatics, University of Colorado Anschutz Medical Campus, Aurora, Colorado, USA

## AUTHOR ORCIDs

Christopher H. Arehart ⓘD http://orcid.org/0009-0001-9575-8329
John D. Sterrett ⓘD http://orcid.org/0000-0002-0931-7181

## FUNDING

| Funder | Grant(s) | Author(s) |
|--------|----------|-----------|
| National Science Foundation (NSF) | | Christopher H. Arehart |
| | | John D. Sterrett |
| | | Ruth E. Quispe-Pilco |
| | | Rosanna L. Garris |
| William J. Freytag Association | | John D. Sterrett |

## AUTHOR CONTRIBUTIONS

Christopher H. Arehart, Conceptualization, Formal analysis, Investigation, Methodology, Project administration, Resources, Software, Validation, Visualization, Writing – original draft, Writing – review and editing | John D. Sterrett, Conceptualization, Formal analysis, Investigation, Methodology, Project administration, Software, Validation, Visualization, Writing – original draft, Writing – review and editing | Rosanna L. Garris, Formal analysis, Investigation, Methodology, Project administration, Validation, Visualization, Writing – original draft | Ruth E. Quispe-Pilco, Formal analysis, Investigation, Methodology, Project administration, Software, Validation, Visualization, Writing – original draft, Writing – review and editing | Christopher R. Gignoux, Conceptualization, Methodology, Project

Research Article | mSystems

administration, Supervision, Validation, Writing – original draft, Writing – review and editing | Luke M. Evans, Conceptualization, Methodology, Supervision, Validation, Writing – original draft, Writing – review and editing | Maggie A. Stanislawski, Conceptualization, Project administration, Supervision, Validation, Writing – original draft, Writing – review and editing

## ADDITIONAL FILES

The following material is available online.

### Supplemental Material

**Supplemental Material (mSystems00677-23-s0001.pdf).** Supplemental figures and tables.

### Open Peer Review

**PEER REVIEW HISTORY (review-history.pdf).** An accounting of the reviewer comments and feedback.

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
