## [Reviewer comments · mSystems]

Poly-omic risk scores predict inflammatory bowel disease diagnosis

Christopher Arehart, John Sterrett, Ruth Quispe-Pilco, Rosanna Garris, Christopher Gignoux, Luke Evans, and Maggie Stanislawski

Corresponding Author(s): John Sterrett, University of Colorado Boulder

Review Timeline:

Submission Date:	July 26, 2023
Editorial Decision:	October 1, 2023
Revision Received:	October 25, 2023
Accepted:	November 2, 2023

Editor: Robert Beiko

Reviewer(s): The reviewers have opted to remain anonymous.

Transaction Report:

DOI: <https://doi.org/10.1128/msystems.00677-23>

September 1, 2023

Dr. John D. Sterrett
University of Colorado Boulder
Department of Integrative Physiology
Boulder, CO

Re: mSystems00677-23 (Poly-omic risk scores predict inflammatory bowel disease diagnosis)

Dear Dr. John D. Sterrett:

Thank you for submitting your manuscript to mSystems. We have completed our review and I am pleased to inform you that, in principle, we expect to accept it for publication in mSystems. However, acceptance will not be final until you have adequately addressed the reviewer comments.

I was on the fence about whether to mark this as "Minor Modifications" or "Reject", as R2's comments are mostly minor. Their concern about biological relevance is warranted, however, and I will need to see more examination of this prior to final acceptance.

Preparing Revision Guidelines

Please return the manuscript within 60 days; if you cannot complete the modification within this time period, please contact me. If you do not wish to modify the manuscript and prefer to submit it to another journal, please notify me of your decision immediately so that the manuscript may be formally withdrawn from consideration by mSystems.

Sincerely,

Robert Beiko

Editor, mSystems

Journals Department
Reviewer comments:

Reviewer #2 (Comments for the Author):

In this resubmission, the authors addressed some of the previous raised issues and refocused the narrative to emphasize their approach. However, there are still some concerns that require further efforts.

- While the authors described they included "metagenomics", "viromics" and "metatranscriptomics" in the manuscript, it appears that all these "3" omics were from one original dataset.(i.e. all of these were extracted from the FASTQ data of the standard shotgun metagenomics sequencing)

In this manuscript, what is termed as "metagenomics" and "viromics" seems to merely represent the taxa identification segment of standard metagenomics. Similarly, what is labeled as "metatranscriptomics" appears to be the functional analysis section of the standard metagenomics. If that is true, I would advise caution in using these terms ("multi-omics," "viromics" ,"metatranscriptomics"), especially when the major finding of this manuscript was "incorporating multiple -omics datasets may enhance prediction."

- 237 microbial species were included in the analysis. It would be good to describe the details of the selection criteria. How many microbial species were identified in total? Incorporating a larger number of microbial species might allow for a more comprehensive utilization of the sequencing data.

- The real biological relevance of the 14 MGN features selected by LASSO in relation to IBD is unclear. None of them were significantly associated with IBD in the multiple regression and all those P value were quite large (seems somewhat dubious) . Which in turn raises concerns about the reliability of the approach used in this manuscript. Any other evidence of biological connection (literature etc.) between these microbes and IBD?

- For the compositional based approaches. I am curious whether we can get comparable results if the reference frames method "songbird" is used.

Editor, mSystems

Thank you for submitting your manuscript to mSystems. We have completed our review and I am pleased to inform you that, in principle, we expect to accept it for publication in mSystems. However, acceptance will not be final until you have adequately addressed the reviewer comments.

I was on the fence about whether to mark this as "Minor Modifications" or "Reject", as R2's comments are mostly minor. Their concern about biological relevance is warranted, however, and I will need to see more examination of this prior to final acceptance.

Reviewer #2 (Comments for the Author):

In this resubmission, the authors addressed some of the previous raised issues and refocused the narrative to emphasize their approach. However, there are still some concerns that require further efforts.

- While the authors described they included "metagenomics", "viromics" and "metatranscriptomics" in the manuscript, it appears that all these "3" omics were from one original dataset.(i.e. all of these were extracted from the FASTQ data of the standard shotgun metagenomics sequencing) In this manuscript, what is termed as "metagenomics" and "viromics" seems to merely represent the taxa identification segment of standard metagenomics. Similarly, what is labeled as "metatranscriptomics" appears to be the functional analysis section of the standard metagenomics. If that is true, I would advise caution in using these terms ("multi-omics," "viromics" ,"metatranscriptomics"), especially when the major finding of this manuscript was "incorporating multiple -omics datasets may enhance prediction."

We thank the reviewer for this cautionary comment. We have ensured that the metagenomics, metatranscriptomics, and viromics data are indeed 3 separate datasets. The Human Microbiome Project 2 (HMP2) methods (<https://doi.org/10.1038/s41586-019-1237-9>) describe how:

- Metagenomic data were generated from sequencing a DNA library
- Metatranscriptomic data were generated via sequencing cDNA libraries from a modified RNAtag-seq protocol
- Viromics data were generated via sequencing viral RNA extracted using the MagMax Viral RNA Isolation Kit.

Thus, we find it appropriate to refer to these data as 3 separate -omics data types from 3 separate library preps, and our analyses are accurately referred to as "multi-omics" in the manuscript.

- 237 microbial species were included in the analysis. It would be good to describe the details of the selection criteria. How many microbial species were identified in total? Incorporating a larger number of microbial species might allow for a more comprehensive utilization of the sequencing data.

There were 578 identified bacterial species in the raw data file downloaded from the HMP2 database. Many of these species, however, had minimal variation across samples. By excluding features that had a standard deviation < 1 , we filtered out features that showed minimal variability across samples and would likely be uninformative in the LASSO analysis. Removing low-variance and low-prevalence features has been recommended by multiple groups (DOIs: 10.1073/pnas.0914005107, 10.1111/1755-0998.13730, 10.1038/nmeth.2276). Thus, 237 of the 578 detected microbial species were carried forward in the analysis.

In order to improve transparency of these methods, we have added a column with the total number of features (before quality control filtering) for each omics dataset to Supplementary Table 1. In this table we now describe 1) the number of identified features in the IBDMDB dataset, 2) the number of features passing QC (missingness and low variation filters), and 3) the number of features selected by LASSO.

Supplementary Table 1. Dataset size and split for training and validation. The table shows the number of samples belonging to each respective category, with the number of participants those samples correspond to shown in parentheses. One participant was removed during the split to training and validation datasets due to missing data. *Because the multi-omic model needed all four -omic data types, only samples that included all four -omic data types could be retained. As a result, samples missing one or more -omics were discarded, and validation samples retained for multi-omic testing are shown in the far right column. Abbreviations: QC, quality control; IBDMDB, inflammatory bowel disease multi-omics database; LASSO, least absolute shrinkage and selection operator; MGN, metagenomics; MTS, metatranscriptomics; VRM, viromics; MBL, metabolomics.

Dataset	N identified features in IBDMDB dataset	N features passing QC (missingness and low variation filters)	N features selected by LASSO	Total samples (participants)	Training samples (participants)	Validation samples (participants)	Validation samples (participants) retained for multi-omic testing*
Metagenomics (MGN)	578	237	14	1627 (130)	1210 (100)	417 (30)	114 (30)
Metatranscriptomics (MTS)	421	280	23	804 (109)	587 (79)	217 (30)	114 (30)
Metabolomics (MBL)	596	269	14	546 (130)	374 (76)	172 (30)	114 (30)
Viromics (VRM)	239	9	6	703 (105)	493 (75)	210 (30)	114 (30)

In the methods we described this quality control process in greater detail with the addition of a sentence (underlined) specifying the initial and remaining number of features per dataset:

“Data processing and thresholding. Compositional datasets (MGN, MTS, VRM) were normalized with center log-ratio transformation, and MBL was normalized using a \log_{10} transformation. We then removed highly sparse features (found in fewer than 5% of samples)

from VRM. For MBL, we only included compounds present in >99% of samples. The difference in methods used for these two data types is due to variation in sparsity of the datasets, as the majority of viruses were found in very few samples, whereas the majority of compounds from MBL were found in most samples. Additionally, we removed one of any two highly collinear features (Pearson's $\rho > 0.95$) from MBL and MTS datasets at random. After normalization, features with a standard deviation less than 1 were excluded from MGN and MTS, and features with a standard deviation less than 0.1 were excluded from VRM and MBL. The standard deviation threshold for each -omics data type was chosen based on a histogram of sample variation in the dataset and served to eliminate features with minimal differences across samples. The exclusion of high-missingness and low-variance features resulted in the filtering of MGN from 578 to 237 features, MTS from 421 to 280 features, VRM from 239 to 9 features, and MBL from 596 to 269 features (see Supplementary Table 1)."

- The real biological relevance of the 14 MGN features selected by LASSO in relation to IBD is unclear. None of them were significantly associated with IBD in the multiple regression and all those P value were quite large (seems somewhat dubious) . Which in turn raises concerns about the reliability of the approach used in this manuscript. Any other evidence of biological connection (literature etc.) between these microbes and IBD?

We'd like to thank the reviewer for this comment, as it has driven some helpful reconsideration of our results. Overall, we'd like to contextualize that MGN was the worst-performing of the 4 omics models, and in the results subsection titled "Individual -omic risk scores" we have specified the inferior performance of the MGN score compared to the other omics:

"The VRM and MBL predicted scores had significant odds ratios of 9.28 [1.53, 56.13] ($p = 0.02$) and 4.30 [1.32, 13.98] ($p = 0.02$) respectively, while MTS had an odds ratio of 2.89 [0.62, 13.52] ($p = 0.2$), and MGN had the lowest odds ratio of 1.24 [0.50, 3.09] ($p = 0.6$)."

Additionally, the model effect sizes for each feature should be considered in a multivariable context and with the limitations of the modeling framework we used. We have highlighted the relatively inferior performance of this taxonomy-based model in the manuscript and expand upon the biological relevance of identified taxa in the discussion.

This is noted in the first paragraph of the discussion section of the manuscript:

"Additionally, our methods allow for comparison of the predictive ability of different -omics for disease, highlighting that taxonomic profiling (from MGN) might not hold as much of a predictive signature of IBD as other -omics"

In order to clarify the context of the discussion, we have added the following subsection to the manuscript as a preface to the discussion of selected features:

"Interpretation of features identified by individual models

In the sections below, we explore the biological relevance of the LASSO-selected features for each of the -omics models. This discussion of the selected features is meant to describe the potential biological relevance underlying each of the models. However, it is important to note that these conclusions are not focused on nominally significant features; instead, the LASSO-selected features are considered in their multivariable context, where many taxa or features could be interdependent (due to factors such as microbial cooperation/competition or mediation of the effects of one pathway by another). We describe features that were selected by LASSO for their joint predictive utility. These should not be confused with results from independent association tests between each feature and the outcome (such as differential expression analysis), which would generally be used to identify individual features significantly associated with the outcome after multiple testing correction. When applied to the testing dataset, the combined multi-omics model was more predictive than the baseline model, and this is the focus of our results; the following sections aim to contextualize the underlying biological relevance of these models.”

To elaborate, the feature p -values from the model should be understood as part of the multivariable analysis, as the effect sizes are the effects of each feature when holding all other features in the model constant. Sometimes these constraints complicate interpretation, as multicollinearity across features often results in high individual feature p -values. Many of these taxa could be interacting as competing or cooperating members of the gut community, resulting in non-linear effects that our regression may not be able to detect. A straightforward example of the effects of multicollinearity on inflating p values for individual feature effects in multiple regression can be seen in the simulated data here: <https://stats.stackexchange.com/a/14528>.

That being said, individual feature p -values could provide some insight into the generality and interpretation of the model, and we understand the reviewer’s comments here. To summarize, we would like to emphasize the following:

1. Our metagenomics model was the least predictive single-omics model (and we highlight that taxonomic information alone may not be sufficient for prediction).
2. The “Interpretation of features identified by individual models” subsection of the discussion aims to describe the potential biological relevance of the LASSO-selected features, which showed predictive utility when validated on a withheld set of samples.
3. When applied to the validation dataset, the combined multi-omics model was more predictive than the baseline model, and this is the focus of our results.

- For the compositional based approaches. I am curious whether we can get comparable results if the reference frames method "songbird" is used.

Our approach using LASSO is similar to the differential ranking approach recommended by Morton et al. in the “reference frames” paper and that is used in Songbird, as we are extracting the features with the strongest effect sizes (i.e., features with small effect sizes are removed from the model). Songbird focuses on the features with the largest relative differentials, like we do in our analysis by only selecting features with the strongest effect sizes in the multivariable model. We agree that Songbird is a great approach for analyzing compositional data; however, it may be redundant to our approach and not well-suited for the metabolomics dataset which is

not compositional. This would complicate the comparison of the relative contribution of each -omics data type to predicting disease status, as it raises questions about which model should be used to analyze the MBL dataset (if not using Songbird) and if it is a fair comparison to use a combination of single-omic scores generated with and without Songbird in the same multi-omics model.

Re: mSystems00677-23R1 (Poly-omic risk scores predict inflammatory bowel disease diagnosis)

Dear Dr. John D. Sterrett:

Your manuscript has been accepted, and I am forwarding it to the ASM production staff for publication. Your paper will first be checked to make sure all elements meet the technical requirements. ASM staff will contact you if anything needs to be revised before copyediting and production can begin. Otherwise, you will be notified when your proofs are ready to be viewed.

Featured Image Submissions: If you would like to submit a potential Featured Image, please email a file and a short legend to mSystems@asmusa.org. Please note that we can only consider images that (i) the authors created or own and (ii) have not been previously published. By submitting, you agree that the image can be used under the same terms as the published article. File requirements: square dimensions (4" x 4"), 300 dpi resolution, RGB colorspace, TIF file format.

Sincerely,
Robert Beiko
Editor
mSystems